# Enhancing Acetic Acid Production in In Vitro Rumen Cultures by Addition of a Homoacetogenic Consortia from a Kangaroo: Unravelling the Impact of Inhibition of Methanogens and Effect of Almond Biochar on Rumen Fermentations

Renan Stefanini Lopes [1,2] and Birgitte Ahring [1,2,3,*]

1   Bioproducts Science and Engineering Laboratory, Washington State University Tri-Cities, Richland, WA 99354, USA
2   Department of Biological Systems Engineering, Washington State University, Pullman, WA 99163, USA
3   The Voiland School of Chemical Engineering and Bioengineering, Washington State University, Pullman, WA 99163, USA
*   Correspondence: bka@wsu.edu

**Abstract:** A homoacetogenic consortium was cultivated from feces from a nursing joey red kangaroo and inoculated into an in vitro ruminal culture. The in vitro ruminal culture was treated with methanogenic inhibitor 2-bromoethanesulfonate (BES), followed by two different homoacetogenic inoculation strategies. Initial observations showed inhibitory effects of BES, with stabilization of the acetic acid concentrations without any increase in concentration, even with the homoacetogenic inoculation. When homoacetogenic bacterial culture was added after the BES addition had ceased, acetic acid production was increased 2.5-fold. Next-generation sequencing showed an increased population of Bacteroidetes after inoculation with the homoacetogenic consortia, along with a slight decrease in diversity. An Almond Shell biochar (AS) addition resulted in a 28% increase in acetic acid concentration if tested directly on the homoacetogenic kangaroo consortia. However, when applied to the rumen culture, it did not enhance acetate production but further promoted other reductive pathways such as methanogenesis and propiogenesis, resulting in increased concentrations of methane and propionic acid, respectively. These findings demonstrate that bioaugmentation with homoacetogenic bacteria can improve acetic acid production of an in vitro rumen culture when methanogenesis has been eliminated. Such advancements can potentially contribute to the optimization of rumen fermentation processes and may have practical implications for improved livestock feed efficiency and methane mitigation strategies.

**Keywords:** homoacetogens; kangaroo; ruminants; methane emissions; almond shell; biochar

## 1. Introduction

The agricultural sector is responsible for almost a quarter of the greenhouse gas (GHG) emissions worldwide, with methane contributing the most to the carbon footprint, corresponding to an estimated 48% of United States agricultural GHG emissions as of 2017 [1].

The primary source of this anthropogenic methane production is ruminant livestock, such as cattle and sheep. These animals have a dedicated fermentation compartment in the stomach (the rumen) where microbes, known as methanogens, thrive and produce methane [2,3]. Methanogenesis is the biological process whereby methanogens produce methane. These microbes belong to the domain archaea using either hydrogen/carbon dioxide, acetic acid, or methanol as substrates. In the rumen, the predominant archaeal genus is *Methanobrevibacter*, growing on hydrogen/carbon dioxide as substrate, which is responsible for consuming the majority of the hydrogen produced during the rumen

fermentation into methane, which is released to the environment by direct release by the animal [4,5].

By utilizing hydrogen as an electron donor, hydrogenotrophic methanogens act as a "hydrogen sink" for excess hydrogen gas, preventing its accumulation. This is vital because a buildup of hydrogen can inhibit the fermentation process and negatively impact rumen microbial populations [6,7]. This beneficial process for hydrolytic bacteria and protozoa makes methanogens a crucial and beneficial part of the rumen's natural microbiota. In fact, most of these archaeal microorganisms are often found closely associated with hydrolytic bacteria and protozoa, where the protozoa, in particular, incorporate these archaea intracellularly. Protozoa can present up to 50% of the total rumen microbial biomass [8,9]. This endosymbiotic relationship makes protozoa an indirect contributor to methane production, for an estimated one-third of methane produced in the rumen, which has been shown to be derived from the endosymbiotic methanogens [10].

In the rumen, methanogens prevail over alternative hydrogen-consuming pathways, possibly due to the fact that methanogenesis is more favorable compared to other processes due to the thermodynamics and further due to the symbiotic relationships that methanogens have developed with protozoa, fungi, and other hydrogen-producing microorganisms [7,9]. Methanogens, however, are not the only microorganisms that are capable of converting electrons created by hydrolytic microorganisms while serving as a hydrogen sink for the anaerobic fermentation systems. Among other hydrogenotrophic microbes, there are bacteria that reduce carbon dioxide into volatile fatty acids (VFAs) using hydrogen as an electron donor. This metabolic process is known as reductive acetogenesis [11].

The diverse group of bacteria capable of autotrophic acetogenesis are called homoacetogens. They utilize acetyl-CoA to produce VFAs via the Wood–Ljungdahl pathway (WLP). Some animals, such as kangaroos, have evolved to explore the energetic advantage of reductive acetogenesis in their enlarged foregut compartment analogous to the rumen [12]. These microbes act much in the same way that methanogens do in the rumen, reducing carbon dioxide by harvesting the redox energy from hydrogen and oxidizing it to water. Just as methanogens, these bacteria work as an electron sink for the kangaroo, increasing the available energy from the ingested biomass [13,14].

When comparing reductive acetogenesis to methanogens in the rumen, the homoacetogenic metabolic route offers not only an alternative to lower methane emissions but also an increase in productivity, as it lessens the gross energy loss associated with methane production. When methanogenesis is inhibited in the rumen, there is not only the availability of hydrogen to be used for reductive acetogenesis but also more available acetic acid, as acetoclastic methanogenesis is inhibited. Though this is not the prevalent form of methanogenesis, acetic acid can be used to produce methane gas by a process known as acetoclastic methanogenesis [15,16]. Overall, acetic acid formed in the rumen of ruminants will be transferred into the bloodstream of the animal and used for its energy needs.

A common target for specific methanogenic inhibition is the methyl-coenzyme M reductase (MCR), responsible for methane synthesis during the methanogenic metabolic pathway [17]. MCR is crucial in all types of methanogens and is a specific enzyme of these microbes.

Chemical structural analogs of MCR, such as 2-bromoethane sulfonate acid (BES), have been shown in studies to effectively reduce methane formation both in vitro and in vivo studies. Resistance to BES has, however, been observed in sheep after successful methanogenic inhibition. Therefore, inhibition with BES will be temporary as the methanogens eventually will gain chemical resistance to the compound [18–21].

Enteric methane inhibition combined with bioaugmentation with homoacetogens could help in sustaining other hydrogen-consuming pathways in the rumen. It might offer a pathway to address two pressing challenges of the current times: agricultural productivity and climate change mitigation. Studies have been performed on the effect of microbial bioaugmentation in rumen cultures, particularly in vitro in semi-continuous bioreactor setups, where improvements in the productivity of acetic acid and other VFAs

were observed [22]. The effects of bioaugmentation were promising, with a substantial increase in acetic acid productivity concurrent with a low hydrogen concentration in the headspace [22–25].

Besides direct chemical inhibition and bioaugmentation, the promotion of electron exchange in the rumen could also benefit thermodynamically unfavorable pathways. This exchange in electrons can happen between microbes, known as interspecies direct inter-species electron transfer (DIET) from hydrolytic to hydrogenotrophic microorganisms. DIET does not require electron carriers; instead, it relies on electron-conductive materials, such as minerals, activated carbon, and biochar [26]. The promotion of DIET through biochar has recently been recognized as an important strategy to improve anaerobic fermen-tation systems and is being studied to be used as a feed additive for the improvement of rumen systems of livestock by regulating the activity of hydrogenotrophic pathways such as methanogenesis and homoacetogenesis [27–30]. Among the array of biochar variants that hold economic appeal for integration into livestock feed additives in the USA is almond shell biochar, a waste from almond production. Almond shells and hulls are already today employed within the dairy industry as bedding and feed additives, respectively [22].

In the present study, we examine the use of a homoacetogenic consortia from a juve-nile kangaroo gastrointestinal (GI) tract collected and cultivated from feces. The effects of bioaugmentation of rumen cultures were analyzed in a series of semi-continuous fermenta-tion in stirred tank reactors simulating the conditions of the rumen using a homoacetogenic culture. As part of these studies, it was further examined the effect of almond shell biochar on acetic acid production by the homoacetogenic kangaroo culture. Biochar is known to be able to change microbial activity and microbial community structures [31]. Recent bioaugmentation in vitro experiments with rumen cultures have demonstrated that model homoacetogens can serve as alternate hydrogen sinks in the rumen while promoting an increase in volatile fatty acid production, but such predominance was found to be only temporary [27,32,33].

This study examines homoacetogenic addition to an in vitro rumen culture along with the introduction of almond shell biochar, which is theorized to promote DIET and improve the performance of homoacetogenic consortia by increasing acetic acid production and consuming more effectively the gas mixture of hydrogen/carbon dioxide and extending the period in which rumen culture remains free of methanogens after being treated once with 10 mM BES.

## 2. Materials and Methods

### 2.1. Biological Collection

Kangaroo fecal samples were collected from Kangaroo Ranch in Fall City, WA, USA. The samples were transported under anaerobic conditions in an ice bath. Fecal samples were collected from a nursing joey red kangaroo (*Macropus rufus*) directly after being dropped in the pouch.

Homoacetogenic culture development was completed by performing dilutional batch fermentation in 120 mL serum vials with 50 mL active volume under 10 PSI pressure and 200 rpm of agitation in an orbital shaker at 37 °C. The initial batch was inoculated with 5% ($w/v$) joey kangaroo fecal samples in 120 mL serum vials with 50 mL of liquid BA medium. The minimum media culture was supplemented with 0.1% yeast extract (YE) along with gaseous substrate addition by pressurizing the headspace of serum vials at 10 PSIA with a gas mixture of hydrogen/carbon dioxide 70:30% or hydrogen/nitrogen 70:30% (Oxarc, Pasco, WA, USA).

Rumen fluid was collected within 2 h after feeding from 2 fistulated 11-year-old Angus cows fed with alfalfa hay by manually collecting solid phase rumen content from its cannula and squeezing it through 2 layers of cheesecloth to collect its liquid content into a 2 L graduated cylinder. The liquid phase collected was transferred to a 2-L reagent bottle that was degassed with 100% nitrogen gas ($N_2$). The collected rumen culture was put in

an ice bath for transportation and sparged with 100% $N_2$ for 10 min before being stored at 4 °C.

### 2.2. Homoacetogens Consortia Batch Fermentation

Fermentation of the homoacetogenic consortia (JK) was performed by batch cultivation in 120 mL serum vials with 50 mL active volume under 200 rpm of agitation in an orbital shaker at 37 °C. BA media was used for serum vial fermentations prepared in accordance with Garret et al. [33]. The vials were purged for 2 min with 100% nitrogen gas ($N_2$) (Oxarc Inc., Pasco, WA, USA) before sealing with rubber stoppers and autoclavation at 120 °C for 20 min. After sterilization, the gas phase was substituted by purging with a gas mixture of carbon dioxide: hydrogen 30/70% (A-L Compressed Gases, Pasco, WA, USA) and was pressurized at 10 psi. The production of acetic acid and gas composition were measured before the consortia were inoculated into the rumen simulator system.

Batch Fermentation with Biochar

Almond shell biochar (AS) pyrolyzed at 475 °C was provided by Corigin Solutions, Inc., Merced, CA, USA. The biochar was ground using an agate stone mortar and pestle (Cole-Parmer Instrument Company, Vernon Hills, IL, USA) and sieved to a particle size below 1.41 mm.

Biochar batch experiment was prepared with BA media as described in 2.2 with the addition of 0.5%, 1%, or 2% ($w/v$) of AS in triplicate sets. Control sets were as described above but without AS addition. Vials were autoclaved at 120 °C for 20 min. After the vials reached room temperature, with the exception of 3 vials used as non-carbon control, the vials were purged using needles with hydrogen: carbon dioxide 70:30% gas mixture for 2 min.

The JK fermentation with biochar was performed in triplicate vials and incubated for 72 h using a 5% inoculum. Gas pressure was measured every 24 h, and the gas phase was flushed and pressurized again to 10 psi using the same gas mixture of carbon dioxide/hydrogen. The final acetic acid concentrations were plotted in RStudio (version 2022.02.1) software programming language for the statistical analysis encompassed two-tailed t-tests to determine whether the means of two groups were statistically different, while analysis of variance (ANOVA) was used for the triplicates using a 95%confidence interval of the mean of concentrations from the cultures with varying biochar concentrations to determine if the mean of measured final acetic acid concentrations were significantly different among the different biochar concentrations. Controls were vials without biochar.

### 2.3. Rumen Simulator Fermentation

Bioreactor systems were used to study the effects of bioaugmentation on the performance of the ruminal culture. The series of bioreactors rumen simulators were set up using a 3 L Applikon® ezControl autoclavable bioreactor (Applikon Biotechnology B.V, Schiedam, The Netherlands) autoclaved at 121 °C for 20 min. After sterilization, each fermenter was filled with 900 mL of rumen fluid under 100% $N_2$ with purging for 15 min. A 5 N sodium hydroxide solution (Sigma Aldrich, St. Louis, MO, USA) was connected to the alkaline controller of the reactor to adjust the pH of the fermentation broth, maintaining a pH of 6.5 throughout the fermentation.

The fermentation was performed at 39 °C, simulating the rumen temperature, and was stirred at a speed of 100 rpm with impellers. The reactors were fed once a day, semi-continuously adding 150 mL of BA media containing 1.0% Avicel, 1.0% starch, and 1.0% yeast extract (Sigma Aldrich, St. Louis, MO, USA). The retention time (RT) of the reactors was 6 days, with the collection of samples prior to feeding. The acclimation period was at least 2 RTs. The addition of 2-bromoethanesulfonate (BES) (Sigma Aldrich, St. Louis, MO, USA) occurred once in each reactor after the acclimation period by adding 2.1 g of BES to the reactor with the feed for a final reactor concentration of 10 mM BES.

Bioaugmented reactors were inoculated once with 10% *v/v* JK homoacetogenic culture during feeding, and the fermentation was run for 72 h under conditions described in Section 2.2.

The setup for the bioreactors is shown in Figure 1.

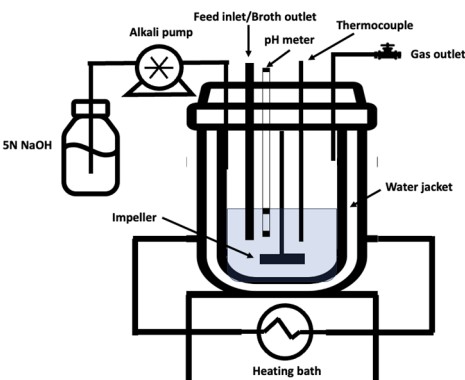

**Figure 1.** Schematic representation of the rumen simulation system.

Rumen Simulation with Biochar

For the rumen simulation with biochar, 2 reactors were run in parallel under the conditions described previously. The acclimation period of the systems was 2 retention times (12 days). The threshold for reactor experiments treatments was set to be 20 days, or 1 RT after biochar addition.

The introduction of biochar to these 2 reactors occurred on day 13 by adding 0.5% *w/v* of almond shell biochar to the feed solution. After a period of 20 days, both reactors were treated once with 10 mM BES on day 30 and bioaugmentation with the JK culture in one of the reactors on day 40. The sampling of both reactors was as previously described.

### 2.4. Analysis of Volatile Fatty Acids

Fermentation broth (2 mL) from the fermenters was centrifuged at 10,000 rpm for 10 min at 4 °C, and the supernatant was analyzed using High-Performance Liquid Chromatography (HPLC). The fermentation broth was diluted 10 times using ddH$_2$O and filtered through a 0.45-micron Nylon filter before being injected and analyzed using an Aminex® HPX-87H Column 250 mm × 4.6 mm (Bio–Rad, Hercules, CA, USA) and a Shodex RI–101 refractive index detector on the UltiMate® 3000 HPLC system (Dionex, Sunnyvale, CA, USA). Sulfuric acid (4 mM) in water was used as the eluent, flowing through the HPX-87H column at a constant flow rate of 0.6 mL/min in a constant temperature oven at 60 °C. The total analysis time of the fermentation sample was 25 min.

### 2.5. Gas Analyses

The gas composition of the headspace was performed by collecting a 10 mL sample from the headspace using a syringe with a Luer-lock valve and injecting it through a peristaltic pump into the Universal Gas Analyzer, UGA Series (Stanford Research Systems, Sunnyvale, CA, USA).

Headspace gas pressure was measured through a rubber connection tube, attached to the rubber stopper, using a bourdon-tube pressure gauge capable of measuring vacuum and adapted with a syringe Luer-lock on its socket.

### 2.6. DNA Isolation

The community of the bioreactors was analyzed after bioaugmentation. DNA isolation from bioaugmented and control rumen samples was performed by centrifuging 10 mL of culture sample at 8000 rpm for 10 min and freeze-drying the cell pellet overnight. The frozen pellet was treated with a bead beater, and 750 μL of CTAB buffer (2%CTAB, 100 mM Tris 8, 20 mM EDTA, 1.4 M NaCl) was added. Vortexed samples were incubated at 65 °C for

one hour, and 300 µL Phenol: Chloroform was added before centrifugation at 14,000 rpm for 5 min. The supernatant was then transferred to a new 1.5 mL Eppendorf tube, and the Phenol: Chloroform extraction was repeated. The final supernatant was then dissolved in an equal amount of 2-propanol and centrifuged at 8000 rpm for 10 min. The supernatant was discarded, and the precipitate was air-dried.

The concentration and purity of the DNA were qualitatively tested using the Nanodrop 1000 spectrophotometer (NanoDrop Technologies, Wilmington, DE, USA) and Qubit 4 Fluorometer (ThermoFisher Scientific, Waltham, MA, USA).

### *2.7. Genomic DNA Sequencing*

Genomic DNA sequencing was performed by Genewiz, Inc. (South Plainfield, NJ, USA) using 16S-EZ library preparation and next-generation sequencing. The extracted DNA was normalized to 20 ng/µL, and the sequencing libraries were prepared using a MetaVX 16S rDNA Library Preparation Kit (Genewiz) and analyzed with an Illumina MiSeq instrument (Illumina, San Diego, CA, USA) according to the manufacturer's instructions. Sequencing was conducted using a 2 × 250 paired-end configuration. The base calls were made by internal Illumina software, and sequence data were demultiplexed and converted to FASTQ format with Illumina's bcl2fastq 2.17 software. Raw digital data were processed using Qiime (version 1.9.1). Pair-end reads were joined, and barcode and primer sequences were removed. Chimeric sequences were removed, and the remaining sequences were clustered into operational taxonomic units (OTUs) against the SILVA 119 database pre-clustered at 97% sequence identity. The Ribosomal Database Program (RDP) classifier assigned a taxonomic category to all OTUs at a confidence threshold of 0.8. Alpha diversity was calculated using both Shannon and Chao1 indices, which are estimators based on abundance.

### 3. Results

#### *3.1. Rumen Simulation Bioreactors without Biochar*

Figure 2 shows the response of the rumen simulation system after treatment with BES without joey kangaroo consortia inoculation. In this bioreactor, hydrogen accumulation occurred during BES treatment, peaking at 34% on day three. The average hydrogen concentration fell after the peak concentration to 13% until day 31, when hydrogen was depleted until methane production picked off again in the system.

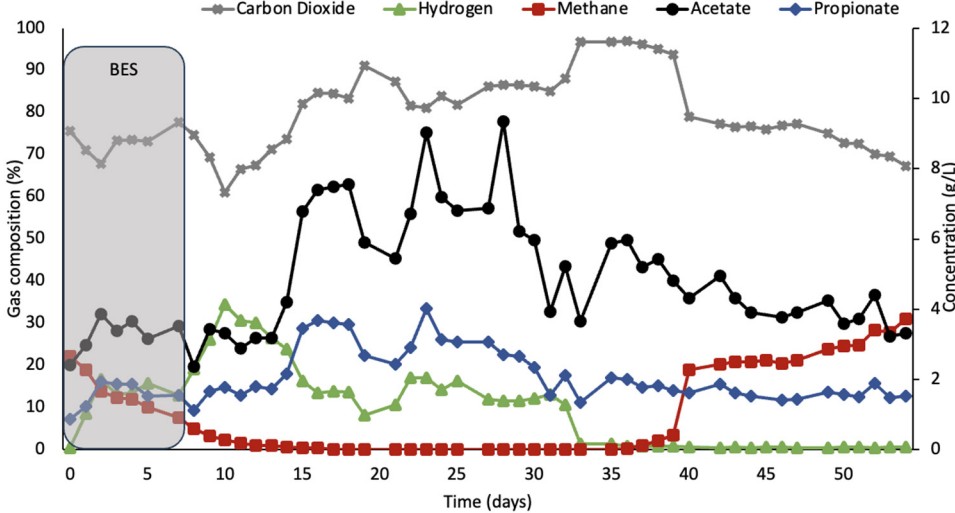

**Figure 2.** Rumen simulation system with 1 RT of 10 mM BES treatment, with a shaded area representing the period when BES was present during one retention time (6 days).

Acetate concentrations were ca. 3.2 g/L during the period of BES treatment and 4.5 g/L after BES addition had been terminated.

As shown in Figure 2, no methane was observed for a total of 20 days, and for 30 days, less than 5% of methane accumulated in the headspace.

Figure 3 shows the rumen simulation system inoculated with homoacetogenic consortia on the third day of 10 mM BES treatment when the first hydrogen peak was observed as a result of BES introduction into the culture. The homoacetogens were introduced into the culture to test for their capability of taking over the role of the methanogens for converting hydrogen and carbon dioxide.

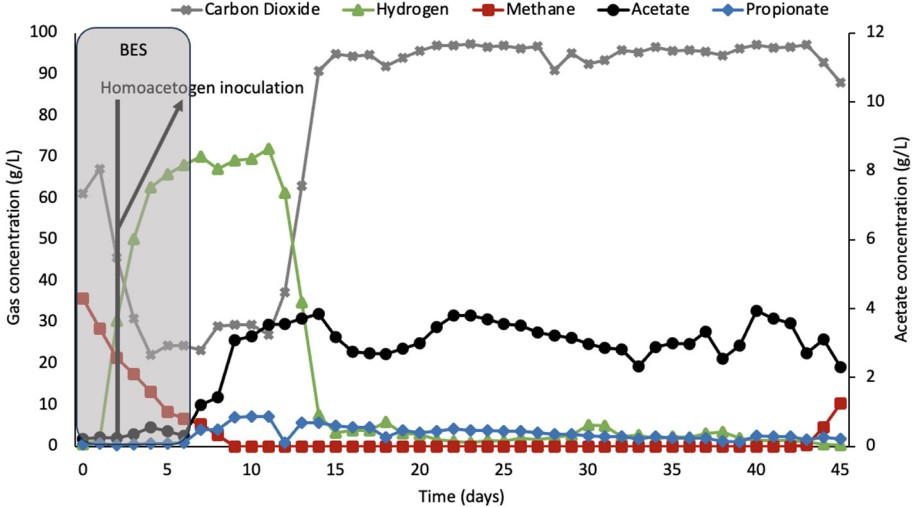

**Figure 3.** Rumen simulation system bioaugmented with kangaroo homoacetogenic culture during 10 mM BES treatment represented as shaded area. The black line indicates the moment of homoacetogenic consortia inoculation.

Hydrogen accumulation started 2 days after BES treatment, and, on day 3, the joey consortia containing homoacetogens were added to the culture. Hydrogen concentrations remained high for a period of 8 days after homoacetogenic inoculation, and no increase in acetate concentrations was observed as well.

The increase in acetate concentrations only occurred after just over one retention time, probably when the BES addition reached a very low concentration, indicating that BES had a direct influence on the homoacetogenic culture. However, the concentration of hydrogen remained high for 5 days after BES treatment was stopped, after which it steadily decreased from 70% to an average of 2.5%. Methane accumulation started to increase again on day 44. Hydrogen concentrations decreased, with the acetate concentrations remaining constant. Hydrogen was fully consumed on day 37, along with the resurgence of methanogens, seen by the accumulation of methane. The acetate average concentration was 3.1 g/L between day 15 and day 43.

The second treatment with BES, shown in Figure 4, was performed where the inoculation of homoacetogenic culture occurred after the BES had again been washed out from the system. As with the other tests, Figure 3 shows that hydrogen concentrations increased and remained steady above 25% during the inhibition period. After it reached a peak of 34.5% hydrogen, the concentration started decreasing 2 days after the addition of the homoacetogenic consortia, and it remained at an average concentration of 3% until small concentrations of methane started to be detected again at day 51.

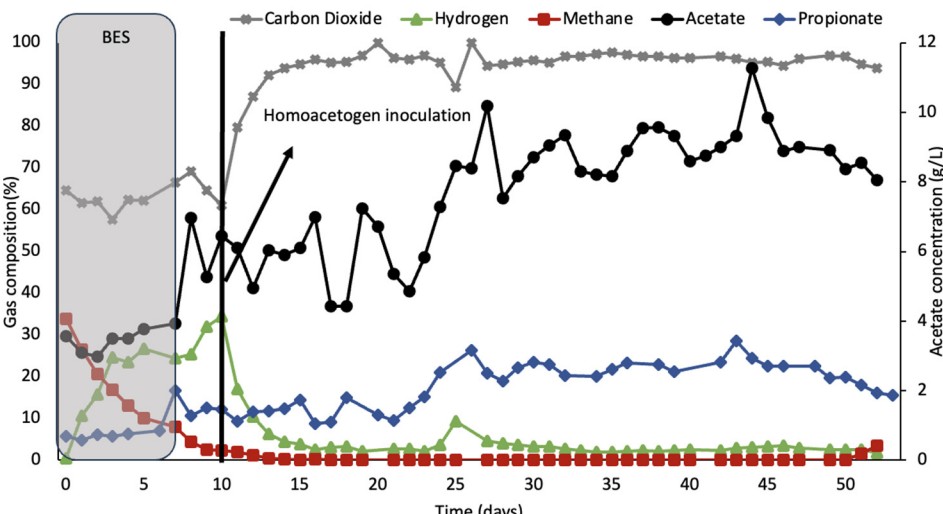

**Figure 4.** Rumen simulation system bioaugmented with kangaroo homoacetogenic culture after 10 mM BES treatment represented as shaded area. The black line on Day 10 represents homoacetogenic consortia inoculation.

Compared to the cultures in Figures 2 and 3, the production of acetic acid in the rumen simulator shown in Figure 4 had 2.4× higher production of acetic acid and maintained a process without methane production for 44 days.

### 3.2. Genomic Studies

The relative abundance of bacterial phyla and genera is presented in Figure 5. The MiSeq V3-V4 reads were classified by the Greengenes 16S rRNA Gene database into 11 phyla, 18 classes, 25 orders, 41 families, and 50 genera.

The reactor bioaugmented with homoacetogenic consortia had a more accentuated increase in OUT pertaining to *Prevotella*, *Sphaerochaeta*, *Butyrivibrio*, *Dysgonomonas*, and *Selenomonas* ($p < 0.05$).

As can be seen in Table 1, the homoacetogenic inoculated culture showed almost no difference in Alpha-diversity (Shannon and Simpson Index) between cultures with and without homoacetogenic inoculation, with a slightly more diverse community in the control reactor without bioaugmentation.

**Table 1.** Alpha diversity of bioreactor culture of Figure 4. The indices for community richness calculation, including ACE and Chao1, were used to estimate the number of OTUs in communities. The indices for community diversity calculation include Shannon and Simpson, both commonly used to reflect the diversity index for the estimation of microbial diversity. Goods Coverage was used to assess the library coverage of each sample. The higher the value, the lower the probability that the sample did not cover the sequence.

| Sample | Ace | Chao1 | Shannon | Simpson | Goods Coverage |
|--------|-----|-------|---------|---------|----------------|
| Con | 1695.447 | 1719.050 | 5.282 | 0.895 | 1 |
| Kan | 1763.928 | 1794.141 | 4.906 | 0.831 | 0.999 |

Con is for the culture without homoacetogenic inoculation. Kan is for the culture with homoacetogenic inoculation.

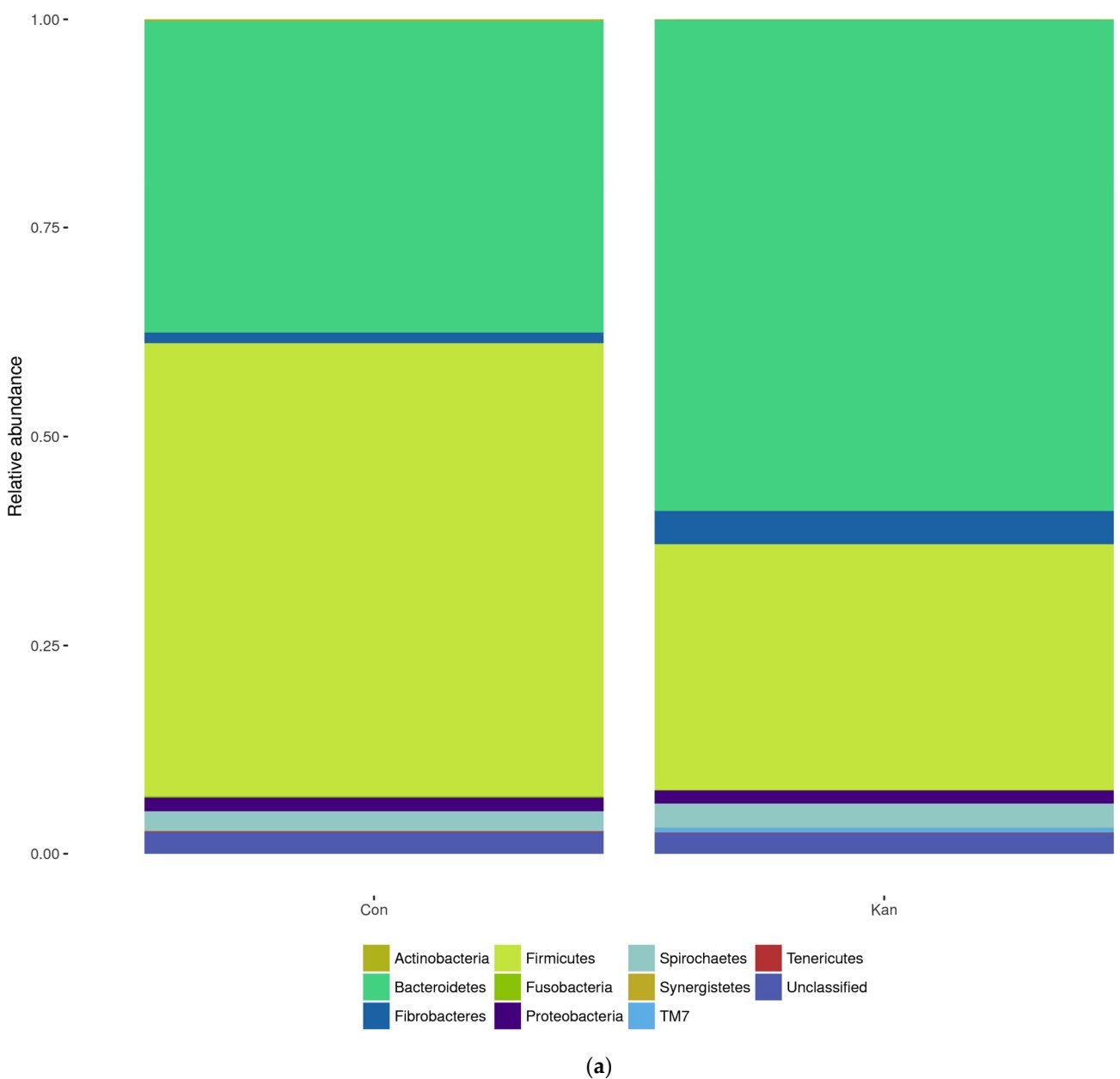

(**a**)

**Figure 5.** *Cont.*

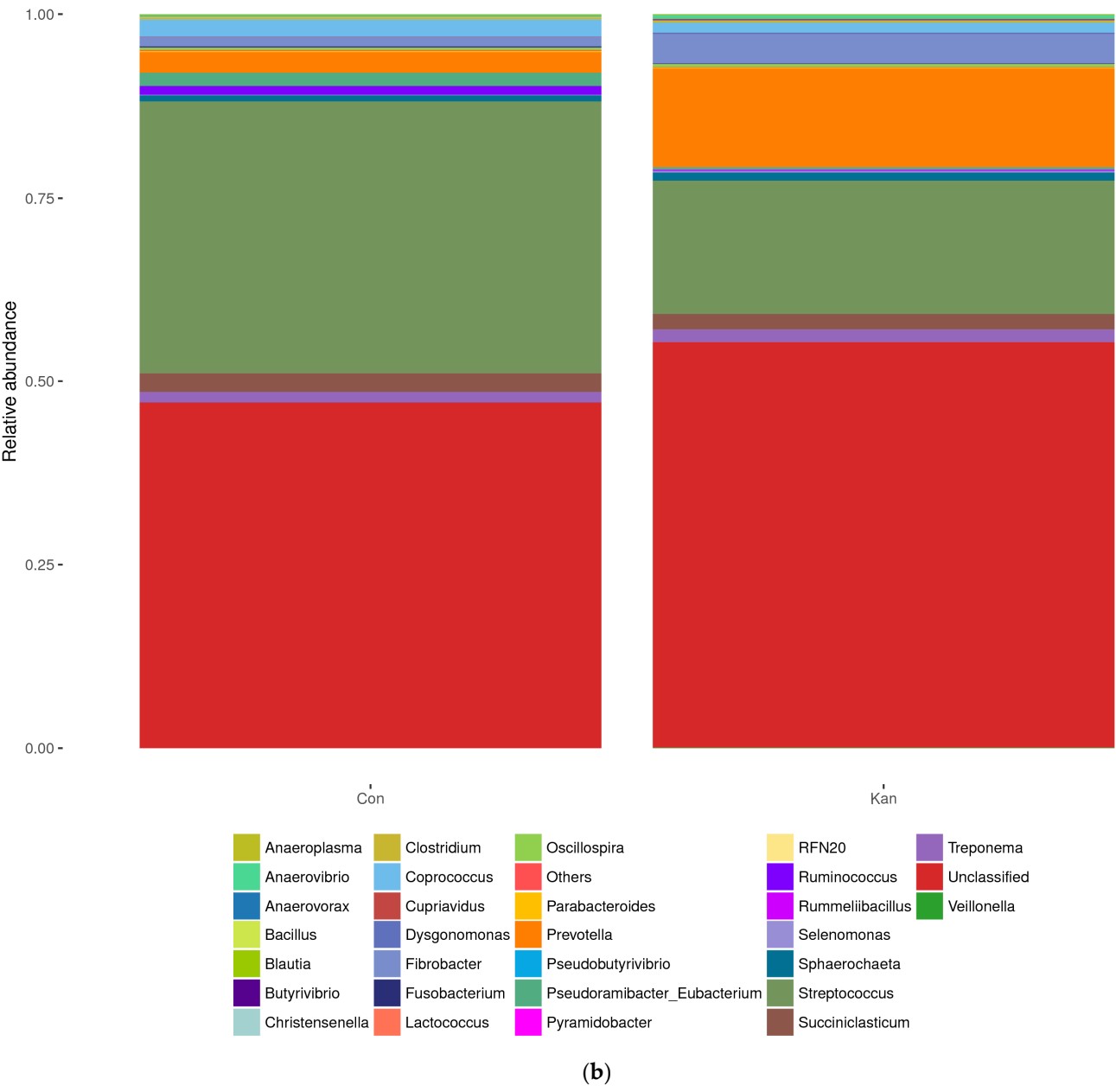

**(b)**

**Figure 5.** Relative phyla abundance of rumen microbial community in the bioaugmented reactor with 16s EZ NGS. Both communities are in vitro rumen cultures treated with BES. "Con" is before bioaugmentation and "Kan" is after bioaugmentation treatment. (**a**) Relative Phyla abundance. (**b**) Relative genus abundance.

### 3.3. Homoacetogen Fermentation with Biochar in Serum Vials

We also tested the effect of AS on the homoacetogenic consortia to see if AS could increase homoacetogenic productivity. We theorize that by increasing the acetic acid productivity, the consortia would be more competitive to methanogens present at the rumen, as the consortia will have increased substrate consumption.

Figure 6B shows the decrease in the headspace pressure of the vials, consisting of hydrogen and carbon dioxide. All of the vials containing almond shell biochar had greater production of acetic acid, in concurrence with what has been observed for the consumption of substrate. The figure indicates the amendment capabilities of biochar regarding the production of acetic acid through reductive acetogenesis, with significantly different means ($p$-value < 0.05) of the final acetic acid concentration at 72 h.

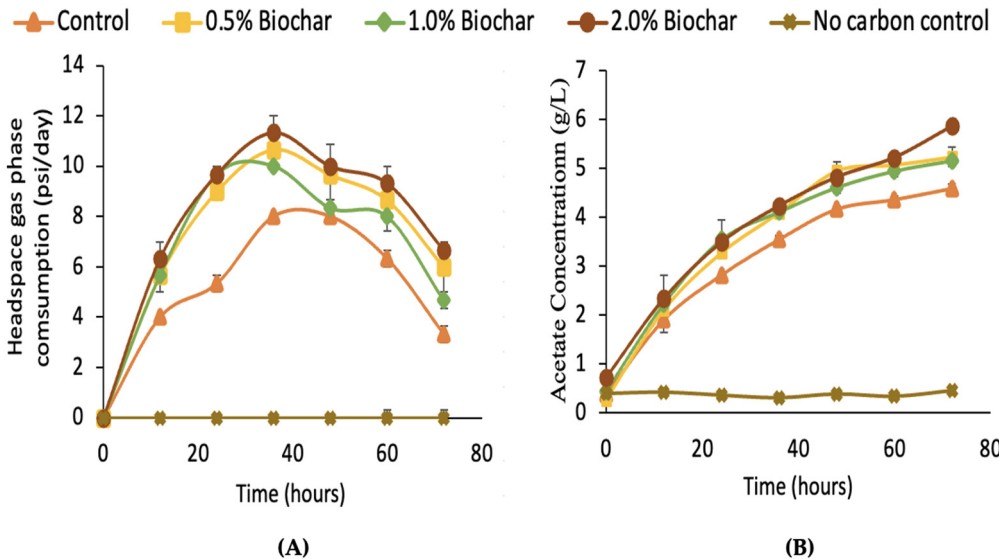

**Figure 6.** Kinetics of homoacetogenic culture from kangaroo feces in serum vials with 50 mL BA media. (**A**) Concentration of acetate in vials with 0%, 0.5%, 1.0%, and 2% *w*/*v* of almond shell biochar (**B**) Average hydrogen: carbon dioxide 70/30% consumption in PSI of with biochar concentrations at 0%, 0.5%, 1.0%, and 2% *w*/*v*. No carbon dioxide control headspace consisted of hydrogen: nitrogen 70/30% and 1% biochar *w*/*v*. *p*-value < 0.05.

After 36 h, substrate consumption peaked. The results show a clear benefit of adding biochar to the homoacetogenic culture, particularly the 2% concentration with the highest production rate for acetic acid, demonstrating its potential to increase homoacetogenesis activity, a desirable trait for enteric methane mitigation.

### 3.4. Rumen Simulation System with Biochar

Figure 7 shows the rumen simulator system where biochar and BES were added with and without bioaugmentation using the homoacetogenic consortia. Figure 7A shows that after the addition of 0.5% *w*/*v* of biochar, the concentration of both propionic acid and methane increased.

Figure 7B shows the rumen simulation system with 0.5% AS biochar with BES treatment followed by bioaugmentation with 10% *v*/*v* JK consortia. As expected, during BES treatment, methane concentrations in the gas phase went down as well as acetic acid. Hydrogen and propionate concentrations increased during BES treatments. Inoculation of homoacetogens occurred on the fourth day after BES treatment, followed by a stabilization of all VFAs.

In Figure 7C, the rumen culture with 0.5% AS biochar was also treated with 10 mM of BES but had no homoacetogenic augmentation. The rumen culture had lower propionate concentrations, indicating a negative correlation between propionate and hydrogen concentrations. For example, on day 49 of the fermentation, hydrogen concentration started to decrease while propionate concentration started to increase.

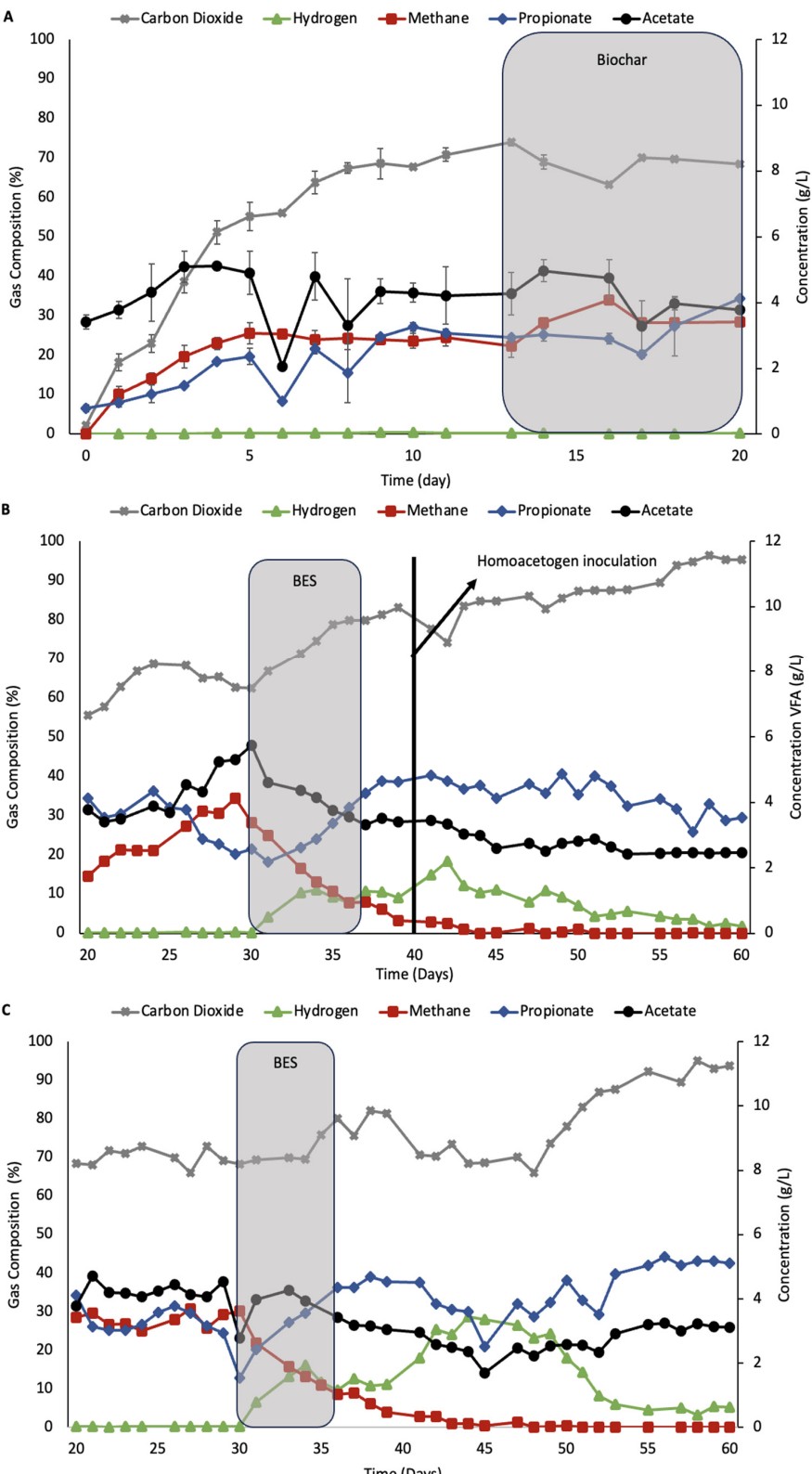

**Figure 7.** Set of 2 rumen simulation systems, with and without kangaroo homoacetogenic culture inoculation. (**A**) Concentrations in the gases and broth during the equilibration period when biochar 0.5% almond shell biochar was added before BES treatments, which is represented by a shaded enclosure. (**B**) Semi-continuous reactor after acclimation period and treated with BES followed by bioaugmentation, represented by a black vertical line. (**C**) Semi-continuous reactor after acclimation period with only BES treatment represented by the shaded region.

## 4. Discussion

The three main VFAs produced in the rumen are acetic acid, butyric acid, and propionic acid. Propionic acid is also known to be a precursor for glucose synthesis in the ruminant's liver through the gluconeogenesis metabolic pathway, and it serves as the principal source of glucose for the animal since very little to no glucose is accumulated in the rumen for absorption [34,35]. Contrary to acetic acid and butyric, propionic acid synthesis can, under certain conditions, occur with the consumption of hydrogen rather than production. Thereby, this process can further act as a hydrogen sink besides methanogens, which generally are the main consumers of hydrogen [36].

The homoacetogenic culture, when used in combination with BES, showed evidence of inhibition, as concentrations of hydrogen remained above 50% throughout the entire BES treatment with no increase in acetate concentrations. This indicates that the homoacetogenic culture is not active during these periods. Only after BES was cleared from the system the hydrogen concentration showed a rapid decrease followed by an increase in acetate concentrations. Previous studies have described the same behavior of inhibition of acetate production even with high hydrogen concentrations after BES additions [22]. BES has also been demonstrated to have a stronger inhibitory effect on bacteria present in the rumen.

Homoacetogens from kangaroos have predominantly been identified as members of the phylum Firmicutes [36,37]. The decrease in their abundance after inoculation in the rumen systems, shown in Figure 5, from 54.19% to 29.37%, was unexpected [37,38]. It is also noticeable that the Bacteroidetes population is more abundant in the bioaugmented culture compared to the non-bioaugmented culture, where Firmicute was the most abundant phylum. Bacteroidetes and Firmicutes are mostly associated with carbohydrate metabolism in the rumen, and other studies have found that both genera are predominantly found in healthy rumen cultures, indicating the robustness of the system after treatments with BES and bioaugmentation [39]. As shown in Figure 4, the concentration of acetate increased to 8.4 g/L after bioaugmentation, an increase of 37% compared to the concentration before the addition of this culture. This was the highest concentration observed in this study. These results are in accordance with previous studies using *Acetobacterium woodii* [22], but our results showed longer periods of time before methane reappeared [22].

The NGS analysis of the rumen simulation system of Figure 5 showed an increase in *Prevotella* and *Butyrivibrio* populations. These are some of the most abundant phyla naturally found in the rumen. This serves as an indicative of a system that resembles the original rumen system, despite the introduction of exogenous bacteria to the rumen, a desirable trade for the use of bioaugmentation [40]. In general, Bacteroidetes are usually the most abundant phyla in the rumen, with numerous studies reporting over 60% relative abundance [41]. Some studies have even linked its predominance with ruminant health indicators [42]. These data, along with the higher observed hydrolytic activity of cellulose and higher production of VFAs, mainly acetate, lead to a propionate ratio closer to three, which strongly indicates that the reactor with bioaugmentation post-BES treatment was able to maintain a healthy ruminal community for over 40 days with homoacetogens taking over the role of the methanogens as an active hydrogen sink.

In the rumen bioreactors with AS biochar treatment, the addition to the rumen culture prior to BES treatments promoted no increase in acetic acid concentrations. However, the concentrations of propionate were increased with a concurrent decrease in the hydrogen concentrations, peaking at 4 g/L propionate. This result strongly suggests that propiogenesis is acting as a hydrogen sink, especially after BES additions in the presence of biochar.

A less pronounced increase in propionate was also observed in Figure 4. The correlated NGS study of Figure 5 shows a noticeable change in the abundance of *Veillonella* spp after bioaugmentation. These microbes are known for their capability to ferment lactate and have been correlated negatively with methane emissions [43]. Lactic acid fermentation to propionate is another hydrogen sink in rumen systems, and studies have been proposed for the manipulation of pathways in the rumen through direct changes in the animal

diet [36,44]. Their higher abundance in the culture without homoacetogens suggests that this phylum could be part of the propiogenesis within the system. The decrease in their relative abundance after bioaugmentation indicates that they might play a less significant role as hydrogen sinks when having to compete with homoacetogens. For instance, when the NGS sample was taken after bioaugmentation in Figure 4, there was a decrease in the concentrations of hydrogen in the culture from day 11 to 20, showing that homoacetogenic populations likely were active and capable of displacing other hydrogen-consuming microbes in the reactor.

All of the different hydrogen-consuming pathways discussed rely on effective electron transport for its functions, and biochar, through its enhancing capability to promote DIET, may favor the most thermodynamic favored pathways, such as methanogenesis and propiogenesis, over less thermodynamically favored pathways such as homoacetogenesis [45,46]. This could explain the lack of acetate production improvement with bioaugmentation, as seen in Figure 7B, where the propionate concentration was 49% higher than the concentration shown in Figure 7A.

The robustness of the rumen system depends on the capacity of its microbial community to be resistant, resilient, and functionally redundant, making it a reliable source for feeding its host. This robustness includes resistance to chemical treatments such as BES [47] or the takeover of functionally similar populations to restore the balance in the system. The results indicate that propiogenesis can play an active role as a hydrogen sink, taking over for the methanogens when needed. Alpha diversity is one of the parameters often used in correlation to rumen robustness, which, in this study, did not significantly change according to Shannon and Simpson indexes (Table 1), showing only a slightly greater diversity in the culture without bioaugmentation [6,48].

The incorporation of AS biochar into the rumen system was beneficial for increased production of acetate by the kangaroo homoacetogenic consortia in the batch fermentations shown in Figure 6, with an improvement of 28% after 72 h of fermentation. The lack of growth in the consortia in the non-carbon dioxide control and sole production of acetate when the consortia were supplied $CO_2$ in the gas phase is a strong indicator that the biochar was promoting improvement in the homoacetogenic activity of the consortia. It has been previously described that biochar can influence the medium pH because of biochar's increased pH buffering capacity, promoting higher bacterial growth [49]. It is noteworthy that the pH for all batches, including the non-biochar control, was ca. 5.7 after 72 h of fermentation, probably due to acetic acid production, showing little interference by the biochar's buffering capacity. The only batch that maintained the initial pH of 6.5 was the non-carbon dioxide control.

## 5. Conclusions

It was demonstrated that a kangaroo homoacetogenic consortia has the capability of maintaining hydrogen concentration low in a rumen system after BES methanogenesis inhibition. Thus, this consortium is capable of serving as the main hydrogen sink in the rumen system, showing that it can outcompete other hydrogen sinks and maintain the system free of methane for over six RTs, lowering the need for recurring chemical treatments for methanogenesis control. However, rumen ruminal microbiota varies with different diets, ages, and animal breeds, making it necessary to further explore this consortia robustness with a greater variety of rumen cultures.

The incorporation of AS biochar pyrolyzed at 475 °C to the homoacetogenic consortia showed an increase in acetate production, with the largest increase observed at 2% $w/v$ of biochar in the culture. The incorporation of 0.5% $w/v$ of biochar in the rumen cultures further showed improvement in methane and propionate yields. Kangaroo homoacetogenic bioaugmentation of rumen in vitro cultures showed no significant change in acetate production, indicating a lack of homoacetogenic activity.

The contrasting data of the biochar regarding acetate production in the rumen simulation system compared to batch vial fermentation can be associated with the strategy adopted in this study for implementation before bioaugmentation.

This study showed that the use of kangaroo homoacetogens for methane mitigation depends on the specific method for its implementation to yield the desired improvement. Therefore, we suggest research to understand the mechanisms at play when amending the rumen culture with almond shell biochar to better promote homoacetogenesis activity in rumen systems.

**Author Contributions:** Conceptualization, R.S.L. and B.A.; methodology, R.S.L. and B.A.; data curation, R.S.L.; writing—original draft preparation, R.S.L.; writing—review and editing, B.A.; supervision, B.A.; project administration, B.A.; funding acquisition, B.A. All authors have read and agreed to the published version of the manuscript.

**Funding:** This research was funded by the WSU CAHNRS Appendix A research program for financial support to Professor B. K. Ahring.

**Institutional Review Board Statement:** Not applicable.

**Informed Consent Statement:** Not applicable.

**Data Availability Statement:** Data are fully available upon request to the corresponding author.

**Acknowledgments:** We thank the Appendix A program for support. We thank the contribution of Kristen Johnson and Jennifer Michal for providing rumen samples. We also thank Lisa Farmen for providing biochar samples, and Rex Paperd from Fall City Wallaby Ranch for providing samples from joey kangaroo.

**Conflicts of Interest:** The authors declare no conflict of interest.

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
