# Peer review of "Enhancing Acetic Acid Production in In Vitro Rumen Cultures by Addition of a Homoacetogenic Consortia from a Kangaroo: Unravelling the Impact of Inhibition of Methanogens and Effect of Almond Biochar on Rumen Fermentations"

_fermentation, doi:10.3390/fermentation9100885_

Round 1

Reviewer 1 Report

There were many careless mistakes, and the diagrams were insufficiently explained, so it was difficult to understand the contents.

Have you received a permission number after deliberation by the Ethics Committee for Animal Experiments?

My specific comments are as follows,

L40,42; H2 or hydrogen ? --- Check the entire text.

L40,42,63; CO2 or carbon dioxide ? --- Check the entire text.

L63, 66; volatile fatty acids --- volatile fatty acids (VFAs)

L81; used for --- The sentence is broken in the middle.

L87; in vitro and in vivo --- Italic or Roman ? --- Check the entire text. for example in L95

L88; sheeps --- OK ? sheep ?

L124; It is not clear what this study will reveal. --- This study investigates the impact of using the methanogenic inhibitor 2-bromoethanesulfonate (BES) and almond shell biochar (AS) on an in-vitro rumen culture bioaugmented with homoacetogenic bacteria derived from kangaroo feces.

L136-201; I think there are too many line breaks.

L147; Batch fermentation with biochar --- What is this sentence?

L148; Corigin Solutions, Inc. --- Where is the company located?

L160; R software --- What is this ? Show us.

L164; concentrations.. --- concentrations.

L192; Rumen simulation with biochar --- What is this sentence?

L223; 8000 --- 8,000

L233; (ThermoFisher Scientific) --- Where is the company located?

L254; the last day of BES treatment --- The last day looks like day 7 in Figure 2 ?

L259; BES was demonstrated to have a stronger inhibitory effect on bacteria present in the rumen, as shown in other studies [24]. --- This is discussion.

L263; Fig. --- Figure

L280; 3 days --- 5 days ?

L277-285; How about carbon dioxide in figure 3 ?

L298-300; This is discussion.

L302; Figure 6 --- Figure 5?

L304; I can't read the letters.

L310; Prevotella, Sphaerochaeta, Butyrivibrio, Dysgonomonas, and Selenomonas --- Italic.

L316-322; Con is for the culture... --- Please transfer to Footnote.

Table 1; 1719.05 --- 1719.050

L345-350; This is discussion.

L352; Figure 5 --- Figure 7

L353; Figure 5A --- Figure 7A

L354; after the addition of 0.5% w/v of biochar --- Did you add biochar during day 13-20 ?

L355-358; Hydrogen accumulation will affect...from glycolysis [22]. --- This is discussion.

L397; Figure 6 --- Figure 5?

L405; inthis --- in this

in Figure 5.. --- in Figure 5.

HAs --- What is this ?

L409; Figure 4 --- Figure 5?

L449; host.. --- host.

L466; volatile fatty acid --- VFA

L473; improvement --- What do you mean ? increase or decrease ?

Figure 2; What is the meaning of the shaded enclosure?

Figure 3; What do the arrows mean? What do the vertical bars mean? What is the meaning of the shaded enclosure?

Figure 4; What do the arrows mean? What do the vertical bars mean? What is the meaning of the shaded enclosure?

Figure 5; The letters are small and difficult to read.

Figure 7; What is the meaning of the shaded enclosure?

References; Isn't the name of the journal abbreviated?

Author Response

Dear Reviewer,

We appreciate you for your precious time in reviewing our paper and providing valuable comments. It was your valuable and insightful comments that led to possible improvements in the current version. The authors have carefully considered the comments and tried our best to address every one of them. We hope the manuscript after careful revisions meets your high standards. The authors welcome further constructive comments if any.

Below we provide the point-by-point responses highlighted in red.

Sincerely

Dr. Birgitte K. Ahring

Professor, Department of Biological System Engineering and Voiland School of Chemical and Bioengineering

Washington State University

Reviewer 2 Report

The manuscript (fermentation-2615276) investigates the effect of selectively inhibiting methanogens using 2-bromoethanesul-fonate (BES) and almond shell biochar (AS) on an in-vitro rumen culture bioaugmented with homoacetogenic bacteria derived from kangaroo feces. Overall, the subject itself is surely worthy of investigation. However, there are several concerns that need to be addressed as follows:

-      Throughout the manuscript, the writing style should be formal from the third-person perspective. Do not use “we” (e.g., in lines 113, 114, 118, … etc) or “our” (e.g., in lines 93, 451, 453… etc ). 

-      The title of the study is too long and could be more concise and direct.

-   In the Abstract, the description of the experimental design is not clearly stated.

-      The introduction section is too long and not focused. Please summarize the introduction section as too many details are given and focus on the study's problem.

-      The hypothesis of the study should be clarified at the end of the Introduction section.

-      Line 128-134: Kangaroo average weight and feeds should be mentioned. The authors should also provide more information about the rumen fluid collection method and the average weight of Angus cows used in the study. 

-      The Materials and Methods (M&Ms) section could be organized by merging the short paragraphs into one or two paragraphs as possible. 

-      The details of statistical analysis should be indicated at the end of the M&Ms section.

-      I cannot find any evidence of statistical analysis, especially in the Figures and Table 1. For example, the standard error value was not provided with each mean, and the P-VALUE was not indicated.

-      The figure legends in the study are very short and lack sufficient detail. They should be more descriptive and able to stand alone, providing readers with a clear understanding of the data presented in the figures. 

-      Line 260: The results section should not include citations. 

-      Line 316-322: The title of the table is too long and should be shortened. Any additional details can be presented in the table footnote. 

-      Lines 462-497: The authors should clearly and concisely summarize the conclusion section without any citations. The authors should also highlight any limitations of the study and potential implications for future research or practical applications.

-

Author Response

Dear Reviewer,

We appreciate you for your precious time in reviewing our paper and providing valuable comments. It was your valuable and insightful comments that led to possible improvements in the current version. The authors have carefully considered the comments and tried our best to address every one of them. We hope the manuscript after careful revisions meets your high standards. The authors welcome further constructive comments if any.

Below we provide the point-by-point responses highlighted in red.

Sincerely

Dr. Birgitte K. Ahring

Professor, Department of Biological Systems Engineering and Voiland School of Chemical and Bioengineering

Washington State University

Reviewer 3 Report

Please add some more information about the Material and Methods as suggested below. Moreover, as the ref 31 has not been published yet is better to include a description of the method

line        

29           Insert kangaroo in the KW as you affirmed that is an important element

81           check there is something missing

132         ref 31 is not available and in any case is preferable to describe the method

133         was the rumen treated in some way? Omogenized? Filtered?

134         Describe the full diet

157         which was the starting inoculum concentration?

164         cancel double punctuation

178         specify semi continuously? Once a day…?

202         which VFA did you analize?

215         which is the method?

259-60   this is discussion

261         use  propionate in the figure

293         fig 4 use concentration (no capital)

298-99   this is discussion

305         the figure is not in place and the legend is too small to read. do not use abbreviations

345-46   this is discussion

360         figure not in place. It is not readable

366         cancel double punctuation

405         cancel double A

409         RSS? Explain acronym

415         add reference to support the statement

449         cancel double punctuation

reference    please double check        

512         journal and pages missing

515         journal missing

524-560-564-566-568-571-576-591-596-601-609-617        pages missing

541         check journal and title 

580         unpublished

599         bold year, cancel p, and complete pages

Author Response

(The authors gave the same response as above.)

Round 2

Reviewer 1 Report

Check again the style of your references.

Figure 7 raises a question. I have written it in red in the attached file.

Author Response

We apologize, there was an issue with a PDF document that included a review track, and Figure 7 was repeated in the text. The manuscript was subsequently corrected to display the correct version of Figure 7. 

Reviewer 2 Report

No further comments are to be addressed.

-

Author Response

We performed editing for English language corrections.